# Primary postpartum haemorrhage and longer-term physical, psychological, and psychosocial health outcomes for women and their partners in high income countries: A mixed-methods systematic review

Su Mon Latt[1]*, Fiona Alderdice[1], Madeline Elkington[1], Mahkawnghta Awng Shar[2], Jennifer J. Kurinczuk[1], Rachel Rowe[1]

1 National Perinatal Epidemiology Unit, Nuffield Department of Population Health, University of Oxford, Oxford, United Kingdom, 2 Health and Nutrition Specialist, United Nations International Children Fund, Myanmar

* sumon.latt@dph.ox.ac.uk

## Abstract

### Objectives

Most research about outcomes following postpartum haemorrhage (PPH) has focused on immediate outcomes. There are fewer studies investigating longer-term maternal morbidity following PPH, resulting in a significant knowledge gap. This review aimed to synthesize the evidence about the longer-term physical and psychological consequences of primary PPH for women and their partners from high income settings.

### Methods

The review was registered with PROSPERO and five electronic databases were searched. Studies were independently screened against the eligibility criteria by two reviewers and data were extracted from both quantitative and qualitative studies that reported non-immediate health outcomes of primary PPH.

### Results

Data were included from 24 studies, of which 16 were quantitative, five were qualitative and three used mixed-methods. The included studies were of mixed methodological quality. Of the nine studies reporting outcomes beyond five years after birth, only two quantitative studies and one qualitative study had a follow-up period longer than ten years. Seven studies reported outcomes or experiences for partners. The evidence indicated that women with PPH were more likely to have persistent physical and psychological health problems after birth compared with women who did not have a PPH. These problems, including PTSD symptoms and cardiovascular disease, may be severe and extend for many years after birth and were more pronounced after a severe PPH, as indicated by a blood transfusion or

**Data Availability Statement:** All relevant data are within the paper and its Supporting Information files.

**Funding:** SL received funding from the Jardine Foundation for her doctoral research studies at the University of Oxford. ME received funding from Oxford Population Health to fund her doctoral research. All other contributions (FA, RR, JK) were undertaken under the auspices of their employment contracts with the University of Oxford. MAS volunteered to be a second reviewer for this study and received no specific funding for their contribution. The funders had no role in study design, data collection, analysis, decision to publish, or preparation of the manuscript.

**Competing interests:** The authors have declared that no competing interests exist.

hysterectomy. There was limited evidence about outcomes for partners after PPH, but conflicting evidence of association between PTSD and PPH among partners who witnessed PPH.

## Conclusion

This review explored existing evidence about longer-term physical and psychological health outcomes among women who had a primary PPH in high income countries, and their partners. While the evidence about health outcomes beyond five years after PPH is limited, our findings indicate that women can experience long lasting negative impacts after primary PPH, including PTSD symptoms and cardiovascular disease, extending for many years after birth.

## PROSPERO registration

PROSPERO registration number: CRD42020161144

## Introduction

Postpartum haemorrhage (PPH), defined by the World Health Organization (WHO) as blood loss of 500 ml or more from the genital tract within 24 hours after birth [1], is the leading cause of maternal mortality and maternal morbidity accounting for 25% of maternal deaths globally [2]. Most maternal deaths due to PPH occur in low and middle income countries [2], but the incidence of PPH is increasing across the high-income countries including Australia, Belgium, Canada, France, the United Kingdom and the USA [3, 4]. According to the Scottish confidential audit of severe maternal morbidity from 2003–2012, PPH makes a significant contribution to major maternal morbidity with a 6% of women with severe PPH requiring peripartum hysterectomy and 12% needing an intensive care admission [5, 6]. Therefore it is important to explore the maternal morbidity as a measure of the burden of PPH in high income settings.

Most research about outcomes following PPH has focused on immediate outcomes including, for example, acute organ failure [7], blood loss, hypovolemic shock and maternal death [8, 9]. There are fewer studies investigating longer-term maternal morbidity following PPH, resulting in a significant knowledge gap around the associations between PPH and subsequent health and wellbeing, including the impact on the psychosocial and emotional wellbeing of women and their partners [8, 10–15]. Identifying longer-term health outcomes following PPH may have implications for policy and clinical care. This could include, for example, increased awareness and advocacy for improved quality of care, and follow-up of women who experienced a PPH and to provide the opportunity for early identification and intervention for any chronic diseases found to be associated with PPH. For example, the association between hypertensive disorders of pregnancy and heart disease is well-established, and pre-eclampsia is now considered to be a female-specific risk factor for cardiovascular disease risk management [16, 17].

A systematic review conducted in 2016 investigated the prevalence of women's emotional and physical health problems following PPH and included six quantitative studies [18]. Several articles investigating potential physical or psychological consequences of PPH have been published since the last systematic search in April 2015 [19–21]. A further systematic review found a potential association between PPH and post-traumatic stress disorder (PTSD), but this

conclusion was limited by the small number of studies investigating this association [22]. Despite a lack of strong conclusions in these reviews, they provide some evidence of persistent physical and psychological problems up to six-month following PPH.

One possible explanation for the heterogeneity in the direction and size of associations found, particularly between PPH and PTSD, is variation in the quality of care received. Quality of care could influence the trajectory of PPH and its acute morbidities as inadequate care can exacerbate the effects of initial haemorrhage [23, 24]. Understanding women's and partners' experience of PPH and their care experiences may therefore help to understand variations in outcomes. Neither of the existing reviews included qualitative research exploring women's experience of PPH and their care, suggesting that a comprehensive systematic review using an integrated mixed-methods approach would be valuable [25].

This systematic review aimed to synthesize, critically appraise, and summarise the evidence about the longer-term physical and psychological consequences of PPH for women and their partners. Women's and partners' experiences of PPH, including care around the time of the PPH, follow-up care and longer-term impacts of PPH, were explored as a secondary objective, with the rationale that these experiences may influence the occurrence of longer-term outcomes.

## Methods

A mixed-methods systematic review was conducted based on the Joanna Briggs Institute (JBI) methodology with modifications for the data extraction and quality assessment tools [26]. The protocol for this review was registered with PROSPERO (CRD42020161144) and the review is reported according to the Preferred Reporting Items for Systematic Review and Meta-Analysis (PRISMA) 2020 checklist (**S1 Checklist**) [27].

### Eligibility criteria

We searched published literature for quantitative cohort, case-control, cross-sectional, or case series studies; qualitative studies including analysis of interviews, free text boxes/open-ended survey questions which explored women's and partners' experiences of care or seeking care and information following PPH, and/or their perceptions about the physical, psychological and psychosocial impact of PPH; and mixed-methods studies with an eligible qualitative or quantitative component.

**Defining PPH.** The definition and the cut off point for the severity of PPH varies [28]. Definitions using various amount of estimated blood loss (EBL), in combination with a post-partum fall in haemoglobin level and specific interventions for PPH have been applied by previous quantitative studies [5, 29, 30]. To ensure maximum sensitivity and allow for variation in the definition of PPH in different studies, the following definition of PPH was used for the inclusion of quantitative studies in this review:

Bleeding from the genital tract, within 24 hours of giving birth, with an estimated blood loss of >500 ml and/or blood transfusion within 24 hours of giving birth and/or having emergency peripartum hysterectomy (EPH) or arterial embolization following PPH, and/or peripartum fall in haemoglobin concentration of > 2g/dl.

Where blood loss was estimated, the method used was not considered as part of the eligibility criteria.

For qualitative studies, about women's and partners experience of PPH and longer-term outcomes, PPH is often more loosely defined, often based on self-report. To maximise

sensitivity and inclusion we included studies which explored experiences of PPH regardless of the definition.

**Defining physical, psychological and psychosocial outcomes.** Physical, psychological and psychosocial health outcomes were defined as any non-immediate and non-fatal physical and psychological health outcomes for women and their partners after PPH including, for example: venous thromboembolism; chronic heart, respiratory, and renal diseases; breastfeeding problems, including breast infections such as mastitis; decline in general health status; psychosocial and emotional wellbeing, including anxiety, depression and post-traumatic stress disorder. Qualitative studies were considered for inclusion if they explored women's and partners' experiences of PPH including their knowledge and perceptions about the health outcomes of PPH, and their care experiences.

**Defining longer term.** In the absence of any universally agreed definition of what is considered to be a 'longer-term' outcome, and to maximise inclusion of post-discharge outcomes that might have an impact on longer-term wellbeing, all physical and psychological health outcomes following PPH after discharge from the hospital were included.

The inclusion/exclusion criteria in Table 1 were applied.

**Table 1. Inclusion and exclusion criteria.**

| Category | Inclusion criteria | Exclusion criteria |
|---|---|---|
| **Population** | Women who gave birth after at least 24 weeks' gestation and had a PPH, and/or their partners. | Women who had a PPH and/or their partners who were a subset of study participants (e.g. women who had a PPH as a subset of women with a range of maternal morbidities) unless separate data for women who had a PPH could be retrieved. |
| **Exposure** | For quantitative studies, primary PPH, defined as: | Quantitative studies where PPH was not the primary exposure of interest or in which PPH was not well-defined and measured by one of the criteria described. |
|  | Bleeding from the genital tract, within 24 hours of giving birth, with an estimated blood loss of >500 ml and/or blood transfusion within 24 hours of giving birth and/or having EPH or arterial embolization following PPH, and/or peripartum fall in haemoglobin concentration of > 2g/dl. |  |
|  | For qualitative studies, experience of PPH regardless of PPH definition. |  |
| **Comparison** | Studies with or without a control or comparison group, comprising participants who did not experience a PPH. | None |
| **Outcomes** | For quantitative studies, any non-immediate and non-fatal physical, psychological and psychosocial health outcomes for women and their partners after PPH. | For quantitative studies: immediate outcomes, including maternal death following PPH, acute organ failure, sepsis, shock, and with intervention outcomes such as hysterectomy, intensive care admission, blood transfusion during initial birth admission; pregnancy-related outcomes, including secondary PPH, or recurrence of PPH in subsequent pregnancies. |
|  | For qualitative studies, women's and/or partners' experiences of care and of seeking care around the time of PPH and during the postpartum period, and their perceptions about the physical, psychological and psychosocial impact of PPH. |  |
| **Time** | For quantitative studies, any follow-up duration after hospital discharge following PPH. | None |
|  | For qualitative studies, all studies regardless of follow-up duration. |  |
| **Setting** | High-income countries as defined by the World Bank [31] at the time of study. | Low/middle-income countries. |
| **Language** | English language | Papers which were not written in English |

## Search strategy

Searches (**S1 File**) were conducted in Medline (Ovid), Embase, Web of Science, CINAHL and PsycINFO on 16th December 2019 and updated on 6th January 2021 and on 31st May 2022, from the inception of each database to the search date. Search strategies were developed for each database using a combination of controlled vocabulary (e.g., 'post-partum/post-natal/peri-partum haemorrhage', "postpartum/postnatal/peripartum haemorrhage", "obstetric haemorrhage", 'maternal morbidity', 'post-natal/postnatal/post-partum/postpartum/peripartum/peri-partum bleeding"), keywords (e.g., "massive bleeding", "transfusion" "hysterectomy") and pre-tested search filters for qualitative research and observational studies [32]. Terms for blood transfusion and hysterectomy were included alongside general morbidity terms to optimise the sensitivity of the search. Within the search strategy, no restrictions were made on language, year of publication or year of study. Exclusion criteria for papers that were not written in English and papers from low and middle-income countries were applied at the study selection stage. Reference lists from all included full text articles were hand searched.

## Screening and study selection

Search results were imported into the reference management software Endnote and duplicates were removed [33]. Search results were then imported into Covidence software for screening and data extraction [34]. Titles and abstracts of all articles identified were independently screened by SL and MS against the inclusion criteria including language eligibility and high-income setting. Full text screening was conducted independently by SL and ME using the above eligibility criteria. Any disagreements were resolved by consensus between reviewers, involving RR when necessary.

## Data extraction

Data extraction was performed independently by SL, and either MS or ME. Study characteristics and outcome information were extracted for both quantitative and qualitative studies using bespoke data extraction forms developed and pilot-tested for this review (**S2 File**). For qualitative studies, all the text and themes reported under findings or results from included studies were imported into NVivo software and treated as a primary textual data source for analysis. Data from mixed method studies were entered in both forms as appropriate.

## Risk of bias and quality assessment

Risk of bias and quality assessment for the included studies was conducted by SM, MA and ME, with two independent reviewers for each study, with any disagreements resolved by consensus as described above. Included quantitative studies (and the quantitative component of mixed-methods studies) were assessed using the risk of bias assessment for non-randomized studies (ROBANS) tool (**S1 Table**) [35].

Quality assessment for qualitative studies was conducted using the Critical Appraisal Skills Programme (CASP) tool [36]. The response options for Questions 1–9 were modified from yes/no/unclear to include the option to respond 'partly' where a paper addressed some items in the checklist associated with the question, but not all. Quantitative and qualitative components of mixed-methods studies were assessed and reported separately.

## Data synthesis

Since the quantitative studies were not homogeneous in terms of study design and outcome measurements, we used a narrative synthesis approach, presenting quantitative study results in

structured tables [37]. Qualitative studies were synthesised using the approach described by Thomas and Harden [38–40]. This involved: 1) free line-by-line coding of the findings of primary studies; 2) the organization of codes in related areas to construct 'descriptive' themes; and 3) the development of 'analytical' themes to answer the research questions [38–40].

We used the JBI convergent segregated approach for mixed-methods systematic reviews, whereby following separate analysis of the quantitative and qualitative evidence, the similarities and differences in findings were compared to produce a narrative summary [26].

## Results

The search yielded 15,742 references, of which 7,625 were duplicates and were removed (**Fig 1**). After removing duplicates, a total of 8,117 references remained. Following title and abstract screening, 127 references were included for full text review. An additional six references were identified through other sources. After removal of ineligible studies, 33 papers reporting 24 studies were included in the review, of which 16 were quantitative, five qualitative, and three mixed-methods studies.

### Characteristics of included studies

Included studies were conducted in Australia, New Zealand, France, the Netherlands, Sweden, Switzerland, South Korea, Canada, the UK and the United States, between 2005 and 2021. The

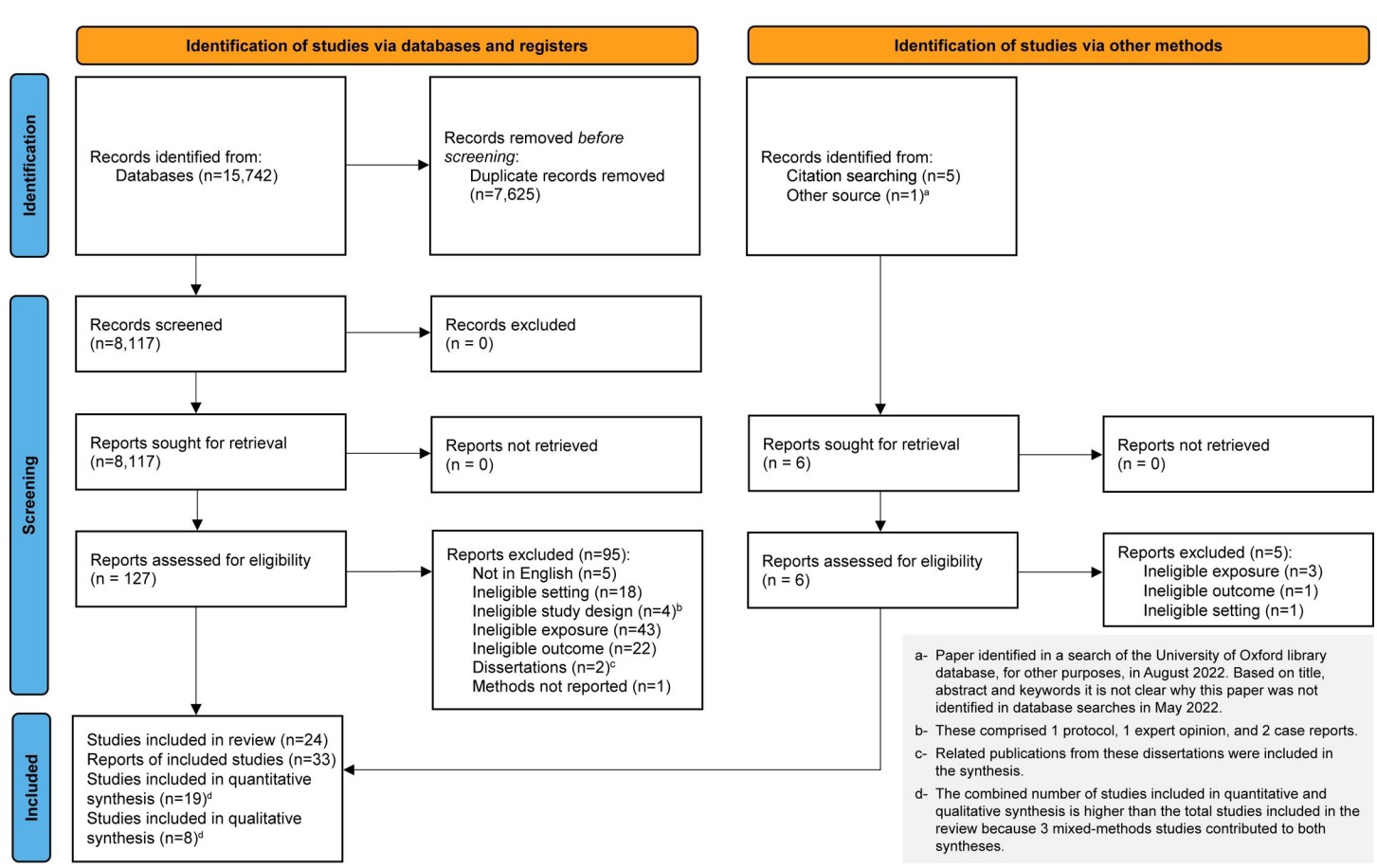

**Fig 1. PRISMA flow diagram for study selection [27].**

follow-up time was from hospital discharge following PPH to up to six weeks postpartum in seven studies [10, 19, 21, 41–45], up to a year in four studies [14, 15, 20, 46–48], up to five years in three studies [49–52], and longer than five years in nine studies [53–64]. One qualitative study did not report on follow-up duration [65]. The definitions for PPH and severe PPH used in these studies varied widely. In studies defining PPH according to blood loss volume, this varied from 500ml to 1000ml for vaginal birth and from 750 ml to 1500 ml for Caesarean birth. In four studies, PPH was identified by using ICD 10 codes recorded in hospital or insurance databases [43, 48, 57, 58]. Definitions for severe PPH also varied widely, from >1000ml to >2000ml. In seven studies where the exposure was severe PPH, severity was also indicated by combining blood loss volume with interventions such as EPH and/or arterial embolization following PPH [46, 50, 53–55, 60, 62]. PPH was not well defined in one qualitative study [45], however, all qualitative studies were included regardless of their definitions as these were not part of the eligibility criteria for qualitative studies.

Summary characteristics of all included studies are presented in Table 2.

## Quantitative studies

**Risk of bias assessment.** The results of risk of bias assessment for quantitative studies and quantitative components of mixed-methods studies are presented in **Table 3**. Most included studies (n = 15) had a low risk of bias (i.e., low risk in most domains assessed). However, four studies had a high risk of bias in two or more areas of domains assessed, with the main sources of potential bias being participant selection, outcome measurement and incomplete data [50, 53, 55, 60].

## Results of quantitative synthesis

Quantitative findings are summarised separately for physical and psychological health outcomes for women and their partners in Table 4.

Two studies reported on general health status in women following PPH [15, 55]. While the first found a decline in general health status at two months and four months following the PPH [15], the second found that there was no difference in SF36 score for women following PPH compared to the SF36 reference group values [55]. Neither study collected data from an appropriate comparison group.

All four studies that explored breastfeeding as an outcome reported that there was no significant difference in the proportion of exclusive breastfeeding at hospital discharge among mothers who had a PPH, compared with mothers who did not have a PPH [14, 41, 42, 46]. However, the proportion of exclusive breastfeeding at hospital discharge (and at six weeks and six months after birth) was slightly lower among mothers who received a blood transfusion, had a low Haemoglobin level or an EPH following PPH, compared with women who did not have a PPH in three studies [41, 42, 46]. One study did not have a comparison group of women who did not have a PPH [14].

Two studies explored the potential association between PPH and venous thromboembolism (VTE) at six weeks postpartum [10, 43]. In one study, severe PPH was strongly associated with the occurrence of VTE in the first six weeks after birth after adjusting for caesarean section and low molecular weight heparin use after birth (OR 5.3, 95% CI: 1.6–17, p = 0.005) [10]. In the second, the authors concluded that PPH was not a major risk factor for VTE unless there was an associated blood transfusion following PPH [43].

Three studies explored the potential association between PPH and the risk of CVD up to 29 years after birth [56–58]. Two studies suggested that women with PPH requiring blood transfusion had a higher risk of hospitalization for CVD (aHR 1.60, 95% CI 1.25–2.06 and aHR 1.38, 95% CI 1.13, 1.68) compared with women without PPH, after adjusting for potential

**Table 2. Characteristics of included studies.**

| Study Country Year of study | Setting | Aims | Population Sample size | Study design | Exposure and definition if provided | Outcome(s) / Themes reported (for qualitative studies) |
|---|---|---|---|---|---|---|
| Quantitative studies | | | | | | |
| Cho 2021 [57] South Korea 2007–2015 | Merged databases of the South Korean insurance claim and health screening programme | To determine whether PPH is associated with cardiovascular disease (CVD) beyond the peripartum period. | Women who gave birth in 2007<br><br>N = 150,381 | Retrospective cohort<br><br>Population-based study | PPH with or without blood transfusion, defined using ICD 10 codes | Newly diagnosed CVD and/or ischemic heart disease |
| Chauleur 2008 [10] France 1999–2004 | University of Nimes Hospital | To explore the effect of severe PPH and its related treatments on postpartum venous thromboembolic risk | Women in their first intended pregnancy<br><br>N = 32,463 | Prospective cohort<br><br>Population-based study | Severe PPH PPH, defined as bleeding occurring in the first 24 hours after birth, persisting after manual exploration of the uterine cavity and requiring I.V prostaglandin administration | Superficial venous thrombosis |
| De la Cruz 2016 [50] United States Year of study not reported | Online community with 1.7 million registered users | To explore if women who had EPH are positive for post-traumatic stress disorder (PTSD) compared with women who did not have EPH more likely to screen | Women from online EPH group<br><br>N = 409 | Retrospective cohort | PPH defined as women who gave birth and had an EPH following PPH | PTSD, postpartum depression (PPD), both PTSD and PPD |
| Chessman 2018 [41] Australia | New South Wales public hospital | To determine the association between red blood cell transfusion and breastfeeding among women who had a PPH at birth taking into account postpartum haemoglobin concentrations | Women who had a PPH and had a red blood cell transfusion following a singleton birth of at least 37weeks' gestation. | Retrospective cohort | PPH with transfusion, defined as blood loss of > = 500ml post-vaginal birth and of > = 750ml post-caesarean birth | Any breastfeeding at hospital discharge |
| | | | Records from 2007–2010 | N = 15,451 | Population-based study | |
| Drayton 2016 [42] Australia Records from 2007–2012 | All New South Wales hospitals | To determine the association between red blood cell transfusion and breastfeeding at discharge among women who had a PPH during their birth admission | All women giving birth in New South Wales, Australia<br><br>N = 40,149 | Retrospective cohort<br><br>Population-based study | PPH and blood transfusion records were identified using diagnosis or procedure codes from the hospital records. Applied ICD10- AM 8th Edition | Exclusive breastfeeding, partial breastfeeding, any breastfeeding |
| Eckerdal 2016 [19] Sweden 2006–2012 | Uppsala University hospital | To explore the association between PPH and PPD | Women who gave birth at University Hospital in Uppsala<br><br>N = 446 | Prospective cohort<br><br>Population-based study | PPH, defined as bleeding of > = 1000ml within 24 hours after birth. | PPD |
| Eggel 2021; Bernasconi 2021 [60, 66] Switzerland Records from 2003–2013 | University Maternity Hospital in Lausanne | To evaluate long term psychological, gynaecological and sexual outcomes among patients with uterine artery embolization following PPH | Women who had PPH and had uterine artery embolization at University Maternity Hospital<br><br>N = 252 women, 142 partners | Case-control<br><br>Hospital-based study | PPH with arterial embolization, defined as women with PPH (>500ml of blood loss) treated with uterine artery embolization | PPD, PTSD, Symptoms of menstrual disturbances, sexual dysfunction |

(*Continued*)

**Table 2.** (Continued)

| Study | Setting | Aims | Population | Study design | Exposure and definition if provided | Outcome(s) / |
|---|---|---|---|---|---|---|
| Country | | | Sample size | | | Themes reported (for qualitative studies) |
| Year of study | | | | | | |
| Feinberg 2005 [51] | A tertiary hospital in Chicago | To determine the incidence of Sheehan's syndrome among patients with obstetric haemorrhage | Women who gave birth at North-western Memorial Hospital with obstetric haemorrhage | Retrospective cohort | PPH, defined as >1000ml blood loss for a vaginal birth or >1500 ml for a Caesarean section | Symptoms of Sheehan's syndrome |
| United States | | | | | | |
| Records from 1998–2002 | | | N = 109 | Hospital-based study | | |
| Knight 2016 [46] | 111 UK hospitals with consultant-led maternity units | To explore the longer-term health outcomes of women following EPH to control haemorrhage and to see how these differ from longer-term health outcomes in women who gave birth without EPH | Women who gave birth and had EPH in UK hospitals | Case-control | PPH with EPH, defined as any woman giving birth and undergoing a hysterectomy in the same clinical episode. | Pain, depression, anxiety, difficulty in sexual intercourse, severe tiredness, menopausal symptoms, flashbacks, difficulty in concentration |
| UK (England) | | | | | | |
| 2013 using records from 2005–2006 | | | N = 78 | Population-based study | | |
| Liu 2021 [48] | Record-linkage using Swedish national birth register linked with national patient register and prescribed drug register | To examine the association between PPH and PPD | Women who had a live birth between 37 weeks and 42 weeks gestational age | Retrospective cohort | PPH, defined as >1000ml blood loss after birth by using ICD-10 code 072 during the birth hospitalization | PPD |
| Sweden | | | | | | |
| Records from 2007–2014 | | | N = 486,722 | Population-based study | | |
| Michelet 2015 [53] | Lariboisiere Hospital in France | To investigate the psychological impact of EPH | Women admitted with PPH who underwent EPH. | Retrospective cohort | PPH with EPH, identified by hospital records of postpartum hysterectomy among PPH patients. PPH was coded as "postpartum complication, haemorrhage shock, or acute anaemia or shock" | PTSD, anxiety, PPD |
| France | | | | | | |
| 2012 using records from 2004–2011 | | | N = 869 | Hospital-based study | | |
| Parry-Smith 2021a; Parry-Smith 2021b [58, 59] | National Health Service (NHS) hospitals in England | To examine long term risk of developing hypertension, CVD and mental ill health among women who had a PPH compared with women who did not | Women aged between 16–46 years who gave births in England | Retrospective cohort | PPH, defined using ICD-10 codes recorded in Hospital Episodes Statistics, England | Hypertension, CVD, PPD, depression, PTSD, anxiety |
| UK (England) | | | | | | |
| Records from 1990–2018 | | | N = 42,327 | Population-based study | | |
| Ricbourg 2015 [20] | Lariboisiere Hospital in France | To investigate the psychological impact of PPH on women and their partners, including its impacts on PTSD, PPD, and the mother/child relationship | Women admitted with PPH | Prospective cohort | PPH: Patients admitted for PPH with a blood loss of > = 500ml, and their partners | PTSD, PPD, mother and infant bonding |
| France | | | N = 40 women, 26 partners | Hospital-based study | | |
| 2010–2011 | | | | | | |

(*Continued*)

**Table 2.** (Continued)

| Study | Setting | Aims | Population | Study design | Exposure and definition if provided | Outcome(s) / |
|---|---|---|---|---|---|---|
| **Country** | | | **Sample size** | | | **Themes reported (for qualitative studies)** |
| **Year of study** | | | | | | |
| Sentilhes 2011 [54] | University affiliated tertiary referral centre at Rouen University Hospital in France | To estimate the longer-term psychological impact of severe PPH | Women with PPH who underwent pelvic arterial embolization at the tertiary obstetric centre | Retrospective cohort | Severe PPH, defined as women who underwent arterial embolization due to PPH. | Psychological symptoms including negative memories |
| France | | | | | | |
| 2009 using records from 1994–2007 | | | N = 68 | Hospital-based study | | |
| Thompson 2010; Thompson 2011 [14, 15] | 17 hospitals across Australia and New Zealand | To explore the physical and psychological health outcomes and breastfeeding outcomes for women following PPH | Women who had primary PPH in Australia and New Zealand | Prospective cohort | Severe PPH, defined as an estimated blood loss of > = 1,500 mL in the 24 hours after child birth, and/or a peripartum fall in haemoglobin concentration | Range of physical and psychological outcomes |
| Australia and New Zealand | | | N = 206 | Population-based study | | |
| 2006–2007 | | | | | | |
| Thurn 2018 [43] | Tertiary hospitals in Sweden | To investigate postpartum blood transfusion and PPH as potential risk factors for postpartum venous thromboembolism (VTE) | Women who gave birth in Stockholm | Retrospective cohort | PPH, defined as an estimated blood loss of > 1000ml within 24 hour after birth | VTE |
| Sweden | | | | | | |
| 2016 using records from 1999–2002 | | | N = 82,376 | Population-based study | | |
| Ukah 2020 [56] | Registry containing all hospital discharges in Quebec, Canada | To investigate the association between obstetric haemorrhage and CVD up to three decades after birth | Women who gave birth in Quebec, Canada | Retrospective cohort | PPH with or without transfusion, defined using ICD 9 and 10 codes and procedural codes for blood transfusion | Hospitalisation for CVD |
| Canada | | | N = 1,224,975 | Population-based study | | |
| Records from 1989–2016 | | | | | | |
| Van Steijn 2019; Van Steijn 2020 [21, 44] | 8 hospitals in Amsterdam, Netherlands | To evaluate whether (i) severe PPH is a risk factor for PTSD in women and (ii) witnessing severe PPH is a risk factor for developing PTSD in partners | Women with ≥ 2000 ml of blood loss and their partners in Netherlands | Prospective cohort | Severe PPH, defined as blood loss > = 2000 ml after birth | PTSD |
| Netherlands | | | N = 308 women, 185 partners | Population-based study | | |
| 2015–2017 | | | | | | |
| Van Stralen 2018 [55] | 98 hospitals with a maternity unit in the Netherlands | To investigate quality of life and psychological impact of major obstetric haemorrhage on women and their partners | Women who experienced major obstetric haemorrhage and had hysterectomy and arterial embolization in the Netherlands | Retrospective cohort | PPH with EPH or embolization, defined as peripartum hysterectomy or embolization after a minimum gestational age of 24 completed weeks | General health status and psychological outcomes |
| Netherlands | | | | | | |
| 2012–2013 | | | N = 58 women, 49 partners | Population-based study | | |
| **Qualitative studies** | | | | | | |
| Briley, 2020 [45] | Postnatal wards in two South London NHS Trusts | To investigate the experience of PPH, for women, birth partners, and health care professionals | Women who had experienced PPH and their partners | Qualitative in-depth-interview study | PPH, definition not provided | Knowledge specific to PPH, effective and appropriate responses to PPH, communication of risk factors, and quantifying blood loss |
| United Kingdom | | | | Maximum variation purposive sampling, semi-structured face to face interviews, thematic analysis | | |
| Not reported | | | N = 9 women, 4 partners | | | |

(*Continued*)

**Table 2.** (Continued)

| Study | Setting | Aims | Population | Study design | Exposure and definition if provided | Outcome(s) / |
|---|---|---|---|---|---|---|
| Country | | | Sample size | | | Themes reported (for qualitative studies) |
| Year of study | | | | | | |
| De La Cruz 2013 [49]<br><br>United States<br><br>Not reported | Online/internet group | To explore women's peripartum experiences of EPH to make recommendations for care | EPH survivors from an English-speaking international internet support group<br><br>N = 15 | Qualitative interview study using grounded theory<br><br>Purposive sampling, semi-structured telephone interviews, constant comparative thematic analysis | PPH defined as women who gave birth and had an EPH following PPH | Death/dying, pain, fear, bonding, numbness/delayed emotional response, communication |
| Dunning 2016 [52]<br><br>United Kingdom<br><br>2013–2014 | One teaching hospital in London | To investigate the experiences of women who have had a primary PPH and the experiences of birth partners who witnessed the PPH | Women who had an estimated blood loss of ≥500 ml within 24 hours after vaginal birth and their partners who were present at the time of the PPH<br><br>N = 11 women, 6 partners | Qualitative interview study<br><br>Maximum variation sampling, explanatory, semi-structured face to face interviews, thematic analysis | PPH, defined as an estimated blood loss of >500ml within the first 24 hour of vaginal birth | Control, communication, consequence (physical and psychological impact), competence |
| Elmir 2012a; Elmir 2012b; Elmir 2012c; Elmir 2014 [61–64]<br><br>Australia<br><br>2009 | University campuses and public places such as pharmacies and child care centres | To describe women's experiences of having an EPH following a severe PPH including impacts on early mothering experiences and to explore the way that women find meaning and positivity in their lives | Women who had an EPH following a severe PPH.<br><br>N = 21 | Qualitative interview study using naturalistic inquiry approach<br><br>Purposive snowball sampling, semi-structured face to face/telephone/e-mail/internet interviews, thematic analysis using inductive method | Severe PPH and emergency hysterectomy, definition not provided | Between life and death, loss of normality, early mothering experiences, moving forward |
| Knight 2016 [46]<br><br>United Kingdom<br><br>2013 | 111 UK hospitals with maternity units | To explore the longer-term health outcomes of women following an EPH to control haemorrhage | Women who had an EPH to control haemorrhage<br><br>N = 78 | Mixed method study using open-ended survey questions<br><br>Purposive sampling, free text questions at the end of follow-up questionnaire, thematic analysis | PPH with EPH, defined as any woman giving birth and undergoing a hysterectomy in the same clinical episode. | The staff providing care, organization / structure of care, longer-term impact of birth experiences |
| Snowdon 2012 [67]<br><br>United Kingdom<br><br>2006–2007 | Two hospitals in UK | To explore women's and partners' experiences of severe PPH and its management | Women who had experienced severe PPH and their partners<br><br>N = 9 women, 6 partners | Qualitative interview study using interpretative phenomenological approach<br><br>Purposive sampling, semi-structured face to face interviews, thematic analysis | Severe PPH, defined by interventions received (uterine tamponade, embolization, laparotomy after vaginal delivery, suture, vessel ligation, or hysterectomy) | Confidence vs fear, trust vs mistrust, satisfaction/ dissatisfaction, communication difficulties |

(*Continued*)

**Table 2.** (Continued)

| Study | Setting | Aims | Population | Study design | Exposure and | Outcome(s) / |
|-------|---------|------|------------|--------------|--------------|--------------|
| **Country** | | | **Sample size** | | definition if provided | **Themes reported (for** |
| **Year of study** | | | | | | **qualitative studies)** |
| Thompson, 2010; Thompson, 2011 [14, 47] | 17 hospitals with maternity services in Australia and New Zealand | To identify sources of distress for women and gaps in service provision, particularly their informational needs | Women who had severe PPH | Mixed-method study using open-ended survey questions | Severe PPH, defined as an estimated blood loss of > = 1,500 mL in the 24 hours after child birth, and/or a peripartum fall in haemoglobin concentration | Adequacy of care, emotional responses to the experience, implications for the future, and concerns for their baby |
| Australia and New Zealand | | | | | | |
| 2006–2007 | | | N = 206 | Purposive sampling, open-ended self-completed survey questions, thematic analysis | | |
| Woiski 2015 [65] | 3 University Hospitals in Netherlands, and childbirth forums on internet | To identify obstacles and facilitators for providing high quality PPH care, from both patient and professional perspectives | Women who gave birth and lost more than 1000 ml of blood after birth | Qualitative interview study | PPH, defined as a blood loss of >1000ml following childbirth | Influencing factors from patient perspectives, professional factors and organizational factors |
| Netherlands | | | N = 12 | Purposive maximum variation sampling, semi-structured face to face/ telephone interviews with patients, framework analysis | | |
| Not reported | | | | | | |

confounders [56, 57]. However, the third study did not find an association between PPH and hypertension or CVD (aHR 1.03, 95% CI 0.87–1.22) [58].

Other physical outcomes investigated, each addressed by one study, included Sheehan's syndrome (damage to the pituitary gland following major haemorrhage) [51], difficulty in sexual intercourse [46], menopausal symptoms [46], menstrual disturbances [60], pain, severe tiredness [46] and fatigue, and mastitis [14]. A higher proportion of women reported these symptoms in severe PPH or EPH groups compared to women who did not have a PPH or EPH.

**Psychological health outcomes for women.** Four studies reported the prevalence of anxiety following PPH at different time points [15, 46, 53, 59]. The exposed group in these studies was women who had a PPH or severe PPH and the follow-up time ranged from two months to eight years after birth. The score for the Hospital Anxiety and Depression Scale (HADS) in the first study indicated moderate anxiety among women who had EPH at median follow-up time of 26.5 months after birth [53]. The second study showed there was mild anxiety among women who had a PPH at both two month and four month follow-up times [15]. Both of these studies did not have comparison groups. The results from the third study indicated women who had hysterectomy were more likely to report anxiety/nerves (27%) compared with women who did not have a PPH (10%) in the first 12 months after birth. Another study, with a follow-up duration of up to 8 years, did not find any association between PPH and anxiety [59].

Nine studies explored potential associations between PPH and postpartum depression (PPD) or depression [15, 19, 20, 46, 48, 50, 53, 59, 66]. with the follow-up time ranging from one month to eight years after birth. Taken together, the findings of these studies were inconclusive, suggesting different directions of association between PPH and depression. Two studies reported a slightly higher prevalence of depression in the PPH group compared with the non-PPH group at one, three and 12 months after birth. [19, 20] and another two studies with a follow-up of up to 8 years found a higher mean depression score among women who had UAE following PPH [66], and a 10% increased risk of developing PPD among women who

**Table 3. Risk of bias for quantitative studies using ROBANS [35].**

| Study | Participant selection | Exposure measurement | Confounding | Outcome measurement | Incomplete data | Selective reporting |
|---|---|---|---|---|---|---|
| Chauleur 2008 [10] | Low | Low | Low | Low | Unclear | Low |
| Chessman 2018 [41] | Low | Low | Low | Low | Unclear | Low |
| Cho 2021 [57] | Low | Low | Low | Low | Low | Low |
| De la Cruz 2016 [50] | High | Unclear | Low | High | Low | Low |
| Drayton 2016 [42] | Low | Low | Low | High | Low | Low |
| Eckerdal 2016 [19] | Low | Low | Low | Low | Unclear | Low |
| Eggel 2021; Bernasconi 2021 [60, 66] | Low | Low | Low | High | High | Low |
| Feinberg 2005 [51] | Low | Low | Low | Low | High | Low |
| Knight 2016 [46] | Low | Low | Low | Low | High | Low |
| Liu 2021 [48] | Low | Low | Low | Low | Low | Low |
| Michelet 2015 [53] | High | Low | High | High | Low | Low |
| Parry-Smith 2021a; Parry-Smith 2021b [58, 59] | Low | High | Low | Low | Low | Low |
| Ricbourg 2015 [20] | Low | Low | Low | Low | High | Low |
| Sentilhes 2011 [54] | High | Low | Low | High | Low | Low |
| Thompson 2010; Thompson 2011 [14, 15] | Low | Low | Low | Low | Low | Low |
| Thurn 2018 [43] | Low | Low | Low | Low | Unclear | Low |
| Ukah 2020 [56] | Low | Low | Low | Low | Unclear | Low |
| Van Steijn 2019; Van Steijn 2020 [21, 44] | Low | Low | Low | Unclear | High | Low |
| Van Stralen 2018 [55] | High | High | High | Unclear | High | Low |

Low-Low risk of bias, High: High risk of bias

had a PPH compared to women who did not have a PPH after adjusting for age, BMI, smoking, ethnicity, birthweight and delivery method (aHR 1.10, 95% CI 1.01–1.21, p 0.037) [59]. The results from multivariable regression models from two further studies suggested no statistically significant association between PPH and PPD at six weeks and 12 months after birth. [19, 48], while a further study reported a slightly higher prevalence of depression in the non-PPH group (6% in PPH vs 9% in non-PPH) at six months after birth [50]. Two studies did not have comparison groups, but one reported a median HAD score of 6 (normal) while the other reported 11% and 13% of women who had a PPH and/or EPH had symptoms of postpartum depression at two and four months after birth [15, 53].

Seven studies explored the association between PPH and post-traumatic stress disorder (PTSD) at different time points ranging from 1 month to 8 years after birth, indicating a potential association between PPH and PTSD [15, 20, 44, 50, 53, 59, 66]. Five studies found a higher risk of PTSD among women who had a PPH and/or UAE or EPH compared with women who did not experience a PPH [20, 44, 50, 59, 66]. Four of these studies also reported adjusted odds (aOR), risk ratios (aRR) or aHR and these studies suggested a significant association between PPH and PTSD up to eight years after birth [aRR = 4.45 (1.5 month); aRR = 2.46 (6 months); aHR = 3.44 (within 12 months); aRR = 1.90 (36 months); aOR = 5.1 (8 years)] [44, 50, 59, 66]. The other two studies did not have comparison groups, but reported respectively 5% prevalence of PTSD among women who had a PPH at two months after birth [15] and 64% prevalence of PTSD among women who had an EPH at the median follow-up of 26.5 months [53].

**Table 4. Physical and psychological health outcomes investigated by included quantitative studies.**

| Study | Sample size (N) | Time since PPH | Outcome | Outcome assessment | Percentage or median/mean (IQR/SD) | Percentage or median/mean (IQR/SD) | Risk ratio (RR) or Odds ratio (OR) or Hazard ratio (HR) |
|---|---|---|---|---|---|---|---|
| | | | | | PPH | No PPH | (95% CI; p value) |
| **Physical health outcomes** | | | | | | | |
| Thompson 2011 [15] | 206 | 2 months | General Health | Median SF36 | 75 (60–80) | NC | NC |
| | | 4 months | | | 72.5 (60–80) | NC | NC |
| Van Stralen 2018 [55] | 58 | 6–9 years | | Mean SF36 | 70 (23.7) | 71.5 | NR (p = 0.23) |
| | | | | | | SF36 reference group | |
| Drayton 2016 [42] | 40,149 | Hospital Discharge | Exclusive breastfeeding | Perinatal database | 81% (PPH only) | 83% | NR |
| | | | | | 71% (PPH with transfusion) | | |
| Chessman 2018 [41] | 15,451 | Hospital discharge | | Perinatal database | 67% | 79% | NR |
| | | | | | (Hb >90g/L) | | |
| | | | | | 71% | | |
| | | | | | (Hb 70-90g/L) | | |
| | | | | | 65% | | |
| | | | | | (Hb<70 g/L) | | |
| Knight 2016 [46] | 78 | 7 days | | Health questionnaire | 39% | 60% | NR (p = 0.089) |
| | | 1.5 months | | | 14% | 40% | NR (p = 0.018) |
| | | 6 months | | | 12% | 19% | NR (p = 0.519) |
| Thompson 2010 [14] | 206 | 7 days | | Health questionnaire | 63% | NC | NC |
| | | 2 months | | | 58% | NC | NC |
| | | 4 months | | | 45% | NC | NC |
| Chauleur 2008 [10] | 32,463 | 1.5 months | VTE | Colour duplex ultrasound | NR | NR | RR = 5.4 |
| | | | | | | | (1.68, 17; p = 0.0047) |
| | | | | | | | aRR = 5.3 |
| | | | | | | | (1.6, 17; p = 0.005) |
| Thurn 2018 [43] | 82,376 | 1.5 months | | ICD 10 diagnostic codes | NR | NR | OR = 2.85 |
| | | | | | | | (1.47, 1.55; p = 0.001) |
| | | | | | | | aOR = 2.7 (1.4, 5.2) |
| | | | | | | | aOR = 1.4 (0.5, 3.5) |
| | | | | | | | (Further adjusted for blood transfusion) |
| Cho 2021 [57] | 150,381 | Up to 8 years | CVD | ICD 10 diagnostic codes | 4.5% | 4.6% | aHR = 0.96 (0.86, 1.07) |
| | | | | | (PPH) | | (PPH) |
| | | | | | 6.9% | | aHR = 1.60 (1.25, 2.06) |
| | | | | | (PPH with transfusion) | | (PPH with transfusion) |
| Ukah 2020 [56] | 1,224,975 | Up to 29 years | | ICD 10 diagnostic codes | NR | NR | aHR = 1.00 (0.95, 1.06) |
| | | | | | | | (PPH) |
| | | | | | | | aHR = 1.38 (1.13, 1.68) |
| | | | | | | | (PPH with transfusion) |
| Parry-Smith 2021a [58] | 42,327 | Up to 8 years | | Diagnostic codes from English primary care database | 0.16% | 0.18% | aHR = 0.86 |
| | | | | | | | (0.52, 1.43; p = 0.57) |

*(Continued)*

**Table 4.** (Continued)

| Study | Sample size (N) | Time since PPH | Outcome | Outcome assessment | Percentage or median/mean (IQR/SD) PPH | Percentage or median/mean (IQR/SD) No PPH | Risk ratio (RR) or Odds ratio (OR) or Hazard ratio (HR) (95% CI; p value) |
|---|---|---|---|---|---|---|---|
| Feinberg 2005 [51] | 109 | 12–40 months | Sheehan's Syndrome | Two or more symptoms in questionnaire | 25% | 7% | NR |
| Knight 2016 [46] | 78 | 12 months | Difficulty in sexual intercourse | Health questionnaire | 27% | 6% | NR |
| | | 8.1 years | | | 13% | 8% | NR |
| Eggel 2021 [60] | 252 | 8.1 years | Sexual dysfunction | Mean FSFI score | 23.8 (0.4) | 23.2 (0.6) | NR |
| Knight 2016 [46] | 78 | 12 months | Menopausal symptoms | Health questionnaire | 17% | 0% | NR |
| | | 8.1 years | | | 33% | 13% | NR |
| Knight 2016 [46] | 78 | 12 months | Pain | Health questionnaire | 43% | 19% | NR |
| | | 8.1 years | | | 10% | 10% | NR |
| Knight 2016 [46] | 78 | 12 months | Severe tiredness and Fatigue | Health questionnaire | 33% | 13% | NR |
| | | 8.1 years | | | 17% | 21% | NR |
| Thompson 2011 [15] | 206 | 2 months | Postpartum fatigue | Median Milligan score | 17 (15–18) | NC | NC |
| | | 4 months | | | 15 (14–17) | NC | NC |
| Thompson 2011 [15] | 206 | 2 months | Mastitis | Health questionnaire | 22% | NC | NC |
| | | 4 months | | | 14% | NC | NC |
| Van Stralen 2018 [55] | 58 | 6–9 years | Patient Perception-Consequences | Mean B-IPQ | 7.5 (2.6) | NC | NC |
| Eggel 2021 [60] | 252 | 8.1 years | Amenorrhoea | Health questionnaire | 12% | 14% | NR |
| | | | Dysmenorrhoea | | 22% | 25% | NR |
| | | | Spotting | | 7% | 7% | NR |
| **Psychological health outcomes** | | | | | | | |
| Michelet 2015 [53] | 869 | 26.5 months | Anxiety | Median HADS | 10 (8–12) | NC | NC |
| Thompson 2011 [15] | 206 | 2 months | | Median STAI scale (State subscale) | 10 (9–11) | NC | NC |
| | | 4 months | | | 10 (9–11) | NC | NC |
| Knight 2016 [46] | 78 | 12 months | | Health and wellbeing questionnaire | 27% | 10% | NR |
| | | 8.1 years | | | 17% | 17% | NR |
| Parry-Smith 2021b [59] | 42,327 | Up to 8 years | | Diagnostic codes from English primary care database | 5% | 5% | aHR = 0.99 (0.90, 1.09; p = 0.881) |

*(Continued)*

**Table 4.** (Continued)

| Study | Sample size (N) | Time since PPH | Outcome | Outcome assessment | Percentage or median/mean (IQR/SD) PPH | Percentage or median/mean (IQR/SD) No PPH | Risk ratio (RR) or Odds ratio (OR) or Hazard ratio (HR) (95% CI; p value) |
|---|---|---|---|---|---|---|---|
| Michelet 2015 [53] | 869 | 26.5months | Postpartum depression / Depression | Median HADS scale | 6 (2–8) | NC | NC |
| Knight 2016 [46] | 78 | 12 months | | Health and wellbeing questionnaire | 33% | 8% | NR |
| | | 8.1 years | | | 13% | 19% | NR |
| Delacruz 2016 [50] | 409 | 6 months | | EPDS (score ≥12) | 6% | 9% | NR |
| | | 36 months | | | 0% | 4% | NR |
| Eckerdal 2016 [19] | 446 | 1.5 months | | EPDS (score ≥12) | 14% | 10% | aOR = 1.81 (0.91, 3.57) |
| Ricbourg 2015 [20] | 40 | 1 months | | EPDS (score ≥12) | 35% | 15% | NR |
| | | 3 months | | | 11% | 0% | NR |
| Thompson 2011 [15] | 206 | 2 months | | EPDS (score ≥12) | 11% | NC | NC |
| | | 4 months | | | 13% | NC | NC |
| Liu 2021 [48] | 486,722 | 12 months | | ICD-10 codes for depression or PPD, or prescription record of antidepressants | NR | NR | aHR = 1.08 (0.99, 1.17) |
| Parry-Smith 2021b [59] | 42,327 | Up to 8 years | | Diagnostic codes from English primary care database | 5.34% | 4.75% | aHR = 1.10 (1.01, 1.21; p = 0.037) |
| Bernasconi 2021 [66] | 142 | 8 years | | MINI score | 4.1 (0.5) | 2.7 (0.3) | NR, (p = 0.015) |
| De la cruz 2016 [50] | 409 | 6 months | Post-traumatic stress disorder | PSS score (≥15) | 9% | 1% | aOR = 2.46 (1.92, 3.16) |
| | | 36 months | | PSS score (≥15) | 24% | 5% | aOR = 1.90 (1.57, 2.30) |
| Michelet 2015 [53] | 869 | 26.5 months | | IESR (≥30) | 64% | NC | NC |
| Ricbourg 2015 [20] | 40 | 1 months | | IESR (≥30) | 45% | 21% | NR |
| | | 3 months | | | 24% | 13% | NR |
| Thompson 2011 [15] | 206 | 2 months | | PCL (>44) | 5% | NC | NC |
| | | 4 months | | | 3% | NC | NC |
| Van Steijn 2020 [44] | 308 | 1–1.5 months | | PCL-5 ≥32) | 7% | 1.7% | RR = 4.45 (0.99, 20.06; p = 0.035) |
| | | 1.5 months | | CAPS-5 | 5.60% | 0% | NR, (p = 0.007) |
| Parry-Smith 2021b [59] | 42,327 | 12 months | | Diagnostic codes from English primary care database | NR | NR | aHR = 3.44 (1.31, 9.03) |
| | | Up to 8 years | | | 0.20% | 0.17% | aHR = 1.17 (0.73, 1.89; p = 0.511) |
| Bernasconi 2021 [66] | 142 | 8 years | | TSQ (≥7) | 3.5 (0.4) | 2.0 (0.2) | aOR = 5.1 (1.5, 17.5; p = 0.001) |
| | | | | | 22.2% | 4.8% | |
| Ricbourg 2015 [20] | 40 | 1 months | Mother-infant bonding | Median Mother Infant Bonding Scale (MIBS) | 1 (0–2) | 0 (0–2) | NR |
| | | 3 months | | | 1 (0–1) | 0 (0–2) | NR |
| Thompson 2011 [15] | 206 | 2 months | General mental health | Median SF36 | 80 (70–80) | NC | NC |
| | | 4 months | | | 80 (70–90) | NC | NC |
| Van Stralen 2018 [55] | 58 | 6–9 years | | Mean SF36 | 75 (17.4) | 75.5 | NR (p = 0.85) |

(*Continued*)

**Table 4.** (Continued)

| Study | Sample size (N) | Time since PPH | Outcome | Outcome assessment | Percentage or median/mean (IQR/SD) | Percentage or median/mean (IQR/SD) | Risk ratio (RR) or Odds ratio (OR) or Hazard ratio (HR) |
|---|---|---|---|---|---|---|---|
| | | | | | PPH | No PPH | (95% CI; p value) |
| Knight 2016 [46] | 78 | 12 months | Difficulty in concentrating | Health questionnaire | 23% | 2% | NR |
| | | 8.1 years | | | 23% | 17% | NR |
| Knight 2016 [46] | | 12 months | Flashbacks to labour or birth | Health questionnaire | 33% | 6% | NR |
| | | 8.1 years | | | 17% | 2% | NR |
| Sentilhes 2011 [54] | 68 | 1–15 years | Negative memories of PPH | Survey question | 68% | NC | NC |
| Sentilhes 2011 [54] | | 1–15 years | Suggestion by hospital staff of a visit to psychologist | Survey question | 66% | NC | NC |

**aRR** = adjusted risk ratio, **aOR** = adjusted odds ratio, **aHR** = adjusted hazard ratio

**NC** = No comparison group, **NR** = Not reported

Outcome measurements:

**SF 36 (Short form health survey):** Scores 0–100 with a higher score reflecting better outcomes

**Milligan score:** Scores 10–14 'no/low' postpartum fatigue, 15–20 'medium', and 21–40 'high' postpartum fatigue

**B-IPQ (Brief Illness Perception Questionnaire):** Scores 0–80 with higher scores reflecting a greater perceived burden of a disease and lower scores reflecting better outcomes

**FSFI (Female Sexual Function Index Questionnaire):** Scores 2–36 with higher scores indicating high sexual activity and satisfaction, and lower scores indicating sexual difficulties or low sexual activity

**HADS (Hospital Anxiety and Depression Scale):** Scores 0–21 with 0–7 normal, 8–10 borderline abnormal, 11-21- abnormal

**STAI (Spielberger State-Trait Anxiety Inventory):** Scores 6–24 with >12 indicating high anxiety

**EPDS (Edinburgh Postnatal Depression Scale):** Scores 0–30 with >12 indicating postnatal depression.

**MINI (Mini International Neuropsychiatric Interview):** Scores 0–10

**PSS Scale (Post-traumatic stress disorder symptom scale):** 17 item checklist with ≥15 indicating PTSD

**IESR Scale (Impact of Events Scale):** Scores 0–88 with ≥30 indicating PTSD

**PCL scale (17 item checklist):** Scores 17–85 with >44 indicating PTSD

**PCL-5 score (20 item self-report tool):** Scores 0–80 with ≥ 32 indicating PTSD

**CAPS-5 score (Gold standard to diagnose PTSD using criterion A, B-E, F, G):** Having at least one B Criterion symptom, one C Criterion symptom, two D Criterion symptoms and two E Criterion symptoms indicates PTSD.

**TSQ (Trauma Screening Questionnaire):** Score 0–10 with ≥7 indicating PTSD

**Mother Infant Bonding Scale:** Scores 0–27 with a high score indicating a disorder related to mother-child bonding.

Physical health outcomes for women.

Other psychological health outcomes investigated included mother-infant bonding [20], general mental health score [15, 55], difficulty in concentrating, flashbacks [46], negative memories of PPH, and suggestion by hospital staff of a visit to psychologist [54]. Results indicated that a higher proportion of women in severe PPH or EPH groups reported having difficulties in concentrations and flashbacks compared with women who did not have a PPH or an EPH.

**Physical and psychological health outcomes for partners.** Three quantitative studies reported on the prevalence of psychological health outcomes among partners of women who had experienced PPH [21, 55, 66]. The first study reported that there was no strong association between PTSD diagnosis and witnessing PPH among partners of women according to a self-report questionnaire [21]. The second study explored the quality of life for partners of women who had experienced severe PPH using SF-36 general health survey tools [55]. The results indicated that partners of women who had severe PPH had better scores overall compared with the reference group's scores from myocardial infarct and systemic lupus erythematosus

patients. In contrast to these studies, the third study reported that there was a higher prevalence of depression and PTSD among partners of women who had PPH compared to partners of women without PPH (11.5% versus 1.5%, p = 0.019) [66].

## Qualitative studies

**Quality assessment.** The results of quality assessment for the eight included qualitative studies and qualitative components of mixed-methods studies are presented in **Table 5**. All but one of the studies had a response of "yes" to at least seven of the questions in the CASP tool, with the remaining study, having a "partially" response for three questions [46]. Only two of eight studies [52, 67] reported the relationship between researchers and participants clearly (i.e., Q6), so it was not possible to assess whether the roles of researchers had a potential impact on the research study and interpretation of findings.

## Results of qualitative synthesis

Analytical themes are presented under four headings: physical impact, psychological impact, psychosocial impact, and experiences of care. **Fig 2** presents the overall structure of the themes arising from the synthesis, organized into four key time periods: during the emergency, immediate hospital recovery, postnatal period up to six months after birth, and longer-term.

**Women's experiences.** *Physical impact*. **Painful recovery.** This theme was contributed to by five studies and encompasses women's experiences of pain at the time of the PPH and for up to a year afterwards [46, 47, 49, 52, 62]. Physical limitations during the immediate postnatal period and beyond were consistently reported by women across five studies [46, 47, 49, 52, 62]. Their descriptions indicated that pain prevented them from performing simple day to day activities such as walking and cooking dinner [52, 62]. One particular concern raised by women was that they were not able to take proper care of their babies as they were not fully recovered from pain [47]. In the longer-term, one woman who experienced an EPH reported that it took a year for her to physically recover [46].

***Obstacles to breastfeeding***. This theme was contributed to by three studies [14, 49, 61]. During their immediate hospital recovery, women reflected that their initial separation from their baby when they were admitted to intensive care was one of the key barriers to initiating breastfeeding [14, 61]. Other key obstacles reported by women included pain, fatigue, limited mobility and posture problems, and delayed milk production [14, 61]. Women also talked about how their efforts and willingness to breastfeed after PPH were not sufficiently supported by health care professionals, including for example being told that they should not be bothered to breastfeed after a traumatic childbirth [14, 49].

*Psychological impact*. **Brush with death.** This theme was contributed to by three studies and encompasses women's acute psychological distress as a result of an unexpected encounter with an emerging life-threatening event [47, 49, 62].

For many women, being so close to death was traumatic and frightening, particularly when women did not anticipate that death was a possibility [47, 49, 62]. Their, overwhelmingly negative, emotional responses to this were described as varying from "shock", to having a "major anxiety attack" or having an "out of body experience" [47, 49, 62]. Elmir et al. (2012) also highlighted that women were afraid of "leaving their young children behind to the unknown" and not knowing "whether or not they would cope without a mother".

***Symptoms of post-traumatic stress and long term recovery***. This theme was contributed to by five studies and describes women's experiences of post-traumatic stress symptoms (PTSD) following PPH [46, 47, 49, 52, 62, 64]. PTSD is characterised by four key symptoms: reliving experiences, avoidance of reminders of trauma, negative thoughts and mood, and hyperarousal [18].

**Table 5. Quality assessment for qualitative studies using CASP tool [36].**

| Study | 1.Clear statement of aims | 2.Qualitative methodology appropriate? | 3. Research design appropriate for the aims? | 4.Recruitment strategy appropriate? | 5. Data Collection addressed the research issue? | 6.Relationship between researcher and participants adequately considered? | 7.Ethical consideration | 8.Data analysis rigorous? | 9. A clear statement of findings? | 10. How valuable is the research? |
|---|---|---|---|---|---|---|---|---|---|---|
| Briley 2020 [45] | Yes | Yes | Yes | Yes | Yes | Can't tell | Yes | Yes | Yes | Moderate |
| De La Cruz 2013 [49] | Yes | Yes | Yes | Yes | Yes | Can't tell | Yes | Yes | Yes | Moderate |
| Dunning 2016 [52] | Yes | Yes | Yes | Yes | Yes | Yes | Yes | Yes | Yes | High |
| Elmir 2012a; Elmir 2012b; Elmir 2012c; Elmir 2014 [61–64] | Yes | Yes | Yes | Yes | Yes | Can't tell | Yes | Partially | Yes | Moderate |
| Knight 2016 [46] | Yes | Yes | Yes | Yes | Partially | Can't tell | Yes | Partially | Partially | Moderate |
| Snowdon 2012 [67] | Yes | Yes | Yes | Yes | Yes | Yes | Yes | Yes | Yes | High |
| Thompson 2010; Thompson 2011 [14, 47] | Yes | Yes | Yes | Yes | Partially | Can't tell | Yes | Yes | Yes | High |
| Woiski 2015 [65] | Yes | Yes | Yes | Yes | Yes | Can't tell | Yes | Yes | Partially | Low |

Items 1–9 scored as Yes, Partially, Can't tell, No

Item 10 scored as low, moderate, or high value

Women described re-experiencing their traumatic birth in the form of vivid memories, flashbacks and nightmares [47, 62]. Triggers for flashbacks included seeing other pregnant women, being in a hospital environment or seeing stories in the media about giving birth [62].

Women also talked about avoiding reminders of trauma. One woman, for example, said that she avoided hospital environments [62]. For some women, reported in two studies, these symptoms were severe, including having suicidal thoughts [46, 62].

While the overwhelming majority of women's observations from most studies were negative, women from one study reflected on being able to take something positive from their experiences, talking about how it helped them find some perspectives and resilience [64].

*Psychosocial impact.* **Interrupted bonding.** This theme was contributed to by three studies and reflects women's perceptions of bonding with their infants and their role as a mother during their recovery in hospital and in the postnatal period [14, 49, 61]. This theme is interlinked with the theme: obstacles to breastfeeding.

Women reported feeling that bonding with their babies was disrupted by physical separation because of their admission to intensive care or their babies' admission to neonatal care. Women talked about having concerns for their babies' wellbeing and not knowing if they were okay [14, 61]. Women reflected that this initial separation prevented them from holding, touching and breastfeeding their babies, and many felt that this was a key barrier establishing a close relationship with their babies.

Women's reports in these studies showed that this initial interrupted bonding could continue in the weeks following hospital discharge for mothers, some of whom experienced

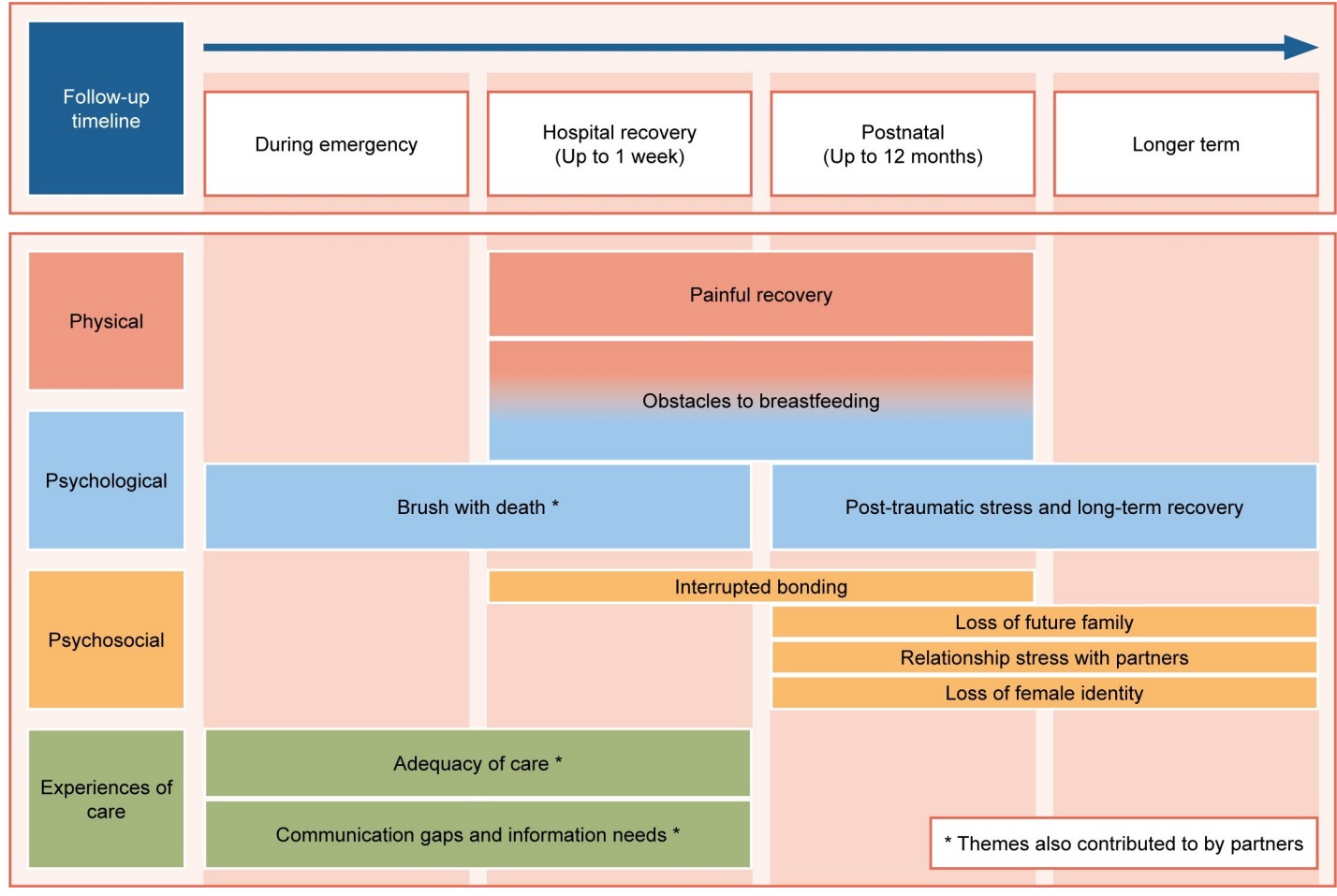

**Fig 2. Summary of analytic themes from thematic synthesis of qualitative studies (n = 8).**

extreme tiredness, continuing for up to a year after childbirth. Some women in one study talked about feeling "upset" and "guilty" when their caring responsibilities were taken away by their family members and partners [61].

*Loss of future family*. This theme was contributed by two studies and encompasses women's feelings of loss in relation to future childbearing following PPH [46, 47]. At four months after childbirth, some women talked about feeling worried about having a PPH in any future births and losing their confidence in their ability to go through birth again [47]. Traumatic birth experiences convinced some women not to have more children [47]. Women who had a peripartum hysterectomy following PPH talked about their regret at not being able to have more children [46].

*Relationship stress with their partners*. This theme was contributed to by three studies and reflects women's perceptions of their relationships with their partner following PPH and/or peripartum hysterectomy, in the immediate postnatal period and up to a year after birth [46, 52, 63]. In one study, one woman who had a PPH described having a relationship breakdown with her partner following a traumatic birth [52]. Women who had a peripartum hysterectomy following PPH talked about sexual tension and intimacy issues with their partners [46, 63]. For instance, a woman in one study mentioned loss of libido and avoidance of physical contact with her partner [46], with this lack of sexual desire and "fear for intimacy" also being echoed

by women in a second study. Women also reported developing insecurities over time causing a strain in their marriage [63].

***Loss of female identity***. This theme was contributed to by two studies and covers women's experiences of peripartum hysterectomy as a consequence of a PPH [46, 63]. Women's accounts reflected that the memories lingered with them for many years and this changed how they felt about themselves as a woman [63]. For some women the lack of uterus was associated with feelings of incompleteness and emptiness [63]. This was echoed by women in the second study as they talked about how peripartum hysterectomy made them feel like a different person and not like a "complete woman" [46].

*Experiences of care*. **Adequacy of care.** This theme was contributed to by six studies. Both positive and negative experiences of care were reported by women, but all demonstrated the crucial role of health care professionals helping women feel calm and supported in an emergency when they established mutual trust [45, 46, 47, 49, 52, 65]. Some women reflected that they appreciated it when the doctors and midwives appeared confident and calm, and took time to reassure them afterwards [45, 52]. Others described feeling unsupported when midwives were "impatient" towards them, "too busy" to explain things or did not "acknowledge" them while providing care. Women's accounts also showed how the care environment could affect their overall experiences. One woman, for example, reported that she was very pleased to be in the critical care unit compared to a busy ward [46].

Insufficient follow-up care and limited resources for psychological health during the postnatal period were identified in four studies [45–47, 49]. Women's accounts indicated a need for psychological counselling and follow-up debrief during the postnatal period [46, 47, 49]. Some women also mentioned that patient information websites and support for breastfeeding would be helpful for them [47, 65].

***Communication gaps and information needs***. This theme was contributed to by six studies and describes women's information and communication needs and preferences, including their information seeking behaviours [45, 46, 47, 49, 52, 65, 67]. Women appreciated when they were given a thorough explanation and clear communication about their condition and treatments received, including hysterectomy and its potential consequences [46]. A lack of information about their current health condition could worsen psychological distress for women and their partners [49, 67].

During the emergency, women appreciated receiving a full explanation and information about their hysterectomy and the indications for it before they consented to have an EPH. Women from five studies consistently reported that they would have liked to know about available treatment options and consequences of PPH, including recovery time after they had experienced PPH [45, 47, 49, 52, 67].

During the immediate recovery period, women's accounts demonstrated that most women were unaware of excessive blood loss [45, 52]. They wanted information about what had happened to them, their current location (e.g. ICU or postnatal ward), and the condition of their babies in simple terms without using confusing technical words [47, 49, 67]. Another communication gap reported by women who had a hysterectomy was that nurses were not aware of their complicated birth and women found it "unsettling" when they had to explain to that they did not have a uterus for examination, and had no need for contraception [49].

Women's information needs extended into the postnatal period, and included communication about the implications of PPH and recovery time, impacts on breastfeeding, and available support services [47, 49, 52, 65].

*Partners' experiences*. Three qualitative studies also included partners of women as participants [45, 52, 67]. Partners' accounts showed that, while women were facing life and death situations, partners also experienced psychological distress from witnessing this [52, 67]. One

partner described himself as "desperate" at being left alone with his baby without support or communication about what had happened to the baby's mother [67]. This was also echoed by others as "distressing", and one partner reported seeking help from the GP and counselling services in the months after witnessing such a traumatic birth [52].

During the emergency and the immediate recovery period, partners appreciated it when doctors took time to reassure them and did not find it helpful when health care staff appeared to show panic or went quiet [45]. In terms of information and communication needs and preferences around the time of the PPH [45, 52, 67], partners wanted more information about available treatment options and potential implications while waiting to find out what had happened [52, 67]. They reported feeling frustrated when they were being asked to leave the emergency room without being fully informed about what was happening [52, 67]. Some suggested that an information sheet about PPH would be helpful for partners and family members to be able to learn more about the situation [67].

## Discussion

The main aim of this review was to identify and summarise the available quantitative and qualitative evidence about longer-term physical, psychological and psychosocial health outcomes for women and their partners following primary PPH in high-income countries. We included 24 studies with a mix of methodological quality ranging from weak to strong. Follow-up time ranged from hospital discharge following PPH to 29 years. Among these, nine studies explored the longer-term impacts of primary PPH after five years, and just two quantitative studies [54, 56] and one qualitative study [61–64] had a follow-up period longer than ten years. Therefore, there is limited research about longer-term health outcomes following primary PPH compared with the many studies that have investigated acute morbidity and health outcomes related to primary PPH [9].

Our review provides evidence that women with PPH are more likely to have persistent physical and psychological health problems during the postpartum period compared with women who have not had a PPH. These differences were more pronounced for women who had some indication of severe PPH such as a blood transfusion, or an EPH. The quantitative findings suggest that women with severe PPH are more likely to have a long recovery time, low general health score, symptoms of tiredness/fatigue, and are less likely to exclusively breastfeed. These quantitative findings are consistent with women's descriptions of their physical functioning and limitations in performing simple day-to-day activities and women's experiences of challenges around initiating and maintaining breastfeeding from qualitative studies. Our review's findings build upon the findings of a previous review conducted in 2016 by including updated literature and by synthesizing available qualitative evidence on women's and partners' experiences of PPH illuminating the different directions and size of effects in quantitative studies [18].

A survey conducted among 372 women who had major obstetric haemorrhage in the Netherlands, in which PPH was not distinguished from an antepartum haemorrhage, indicated that 28% of the participants experienced longer-term negative impacts up to six years after birth [68]. Major obstetric haemorrhage was found to have a negative impact on the partner and family, and on their work, and 25% of these women reported having an additional absence from work in addition to maternity leave [68].

Bearing in mind the limitations identified in terms of the quantity and quality of the evidence, our review indicates that some longer-term physical and psychological health outcomes after PPH may be severe and extended several years after childbirth. Included quantitative studies indicated that the prevalence of PTSD symptoms ranged from 3% to 64% among

women with PPH. Women's descriptions from qualitative studies, which included key symptoms of PTSD such as flashbacks and nightmares of having PPH, were supportive of these quantitative findings. These PTSD symptoms were commonly present among women who had severe PPH and lasted for years after birth. This finding is supported by qualitative studies which concluded that childbirth related PTSD can have severe and lasting effects on women and their relationships with family [69–71]. In the general population, the mean prevalence of postpartum PTSD is 4%-6% and may be higher, with a mean prevalence of 19% among women with physical complications and mental health problems such as depression [72, 73]. While our review's findings are indicative of a higher prevalence of PTSD symptoms following primary PPH [20, 44, 50, 66], more evidence is required before concluding a causal association between PPH and PTSD [22].

Our review included three studies which explored the association between PPH and CVD, and the findings suggested that women who had a severe PPH requiring a transfusion were more likely to develop CVD in later life. Prior studies have suggested that women with severe PPH are at a higher risk of acute heart failure and myocardial ischaemia which results from haemorrhagic shock [74, 75]. Therefore, there is a potential that this acute failure has implications for the cardiovascular system which may put women at higher risk of developing subsequent CVD in later life. Other obstetric complications, including pre-eclampsia, preterm birth, maternal anaemia, and gestational diabetes, have also been shown to increase the subsequent risk of developing CVD [76–79]. It is therefore important to consider women's obstetric history before reaching conclusions about a casual association between PPH and CVD.

Qualitative evidence included in this review indicated that whether women received adequate support during the immediate recovery and postnatal period may affect their acute psychological wellbeing and have an impact on longer-term psychological health outcomes after PPH. For example, women's accounts suggested that they received insufficient follow-up care and had unmet needs for breastfeeding and psychological health services. Health care professionals and family members were crucial in helping women feel calm and supported. This is supported by qualitative studies conducted in Australia which have described how care providers' interactions and postnatal support from partners can affect women's experiences of trauma during childbirth [80–82].

We found very limited evidence about outcomes for partners after PPH, with only three quantitative studies without comparison groups. The evidence from one study suggested an increased risk of PTSD among partners who witnessed PPH while another two studies did not indicate any strong associations between PTSD or general health status. However, partners' accounts in qualitative studies indicated that they experienced acute psychological stress and anxiety due to inadequate information when women were in the emergency room, and some of them required counselling services after witnessing PPH. PPH may also have an impact on relationships between women and their partners. Witnessing their partner having PPH and having relationship difficulties may affect partner's psychological wellbeing beyond the postnatal period. Qualitative studies into fathers' experiences who witnessed birth trauma have identified that this can affect their mental health and relationships with friends and family long into the postnatal period [83, 84]. Studies have also found that men are more likely to cope with these traumatic experiences by avoidance, and may be reluctant to seek necessary care which may lead to poor psychological health outcomes [85, 86]. Therefore, it is crucial to investigate the potential psychological impact among partners who have witnessed a PPH.

## Strengths and limitations

To our knowledge this is the first mixed-methods systematic review integrating quantitative and qualitative evidence to explore the potential longer-term health outcomes of primary PPH. The inclusion of qualitative evidence about women's and partners' experiences helped illuminate our understanding of findings from the quantitative studies. A comprehensive and reproducible search strategy was conducted, the methods were rigorous, and quality assessment and data extraction were performed by two authors independently of each other.

Many of the limitations of this review reflect shortcomings of the included studies. First, there were some challenges associated with drawing conclusions about the association between PPH and some health outcomes because most included studies did not have appropriate comparison groups and conclusions were made from the prevalence/proportions among the exposed group without consideration of potential confounders. For most studies, the follow-up time was less than five years. Most studies used self-reported health questionnaires, rather than objective measurements, such as clinical diagnosis.

At the review level, the heterogeneity of studies, including different definitions of PPH, different follow-up time and variation in the outcomes investigated, meant that it was not possible to conduct meta-analysis. Additionally, the number of studies that made an attempt to establish the potential longer-term impact of PPH was limited. We explicitly chose to focus on research from high-income countries because of identified increasing incidence of PPH and because the review was carried out as part of a larger project exploring longer-term outcomes of PPH in a UK population. Given the significant contribution of PPH to maternal morbidity in low and middle income countries, there would be value in a systematic review of longer-term outcomes in those settings. Due to resource limitations, we were unable to consider for inclusion five potentially eligible papers that were not written in English. Furthermore, we were unable to search the grey literature and reports from WHO, United Nations Population Fund, NHS maternity statistics as we originally planned in the Prospero protocol.

## Implications for research

There is limited evidence about the longer-term impact of primary PPH on women and their partners, hence there is a need for further research in this area. In particular, cohort studies with comparison groups, and longer duration of follow-up should be conducted to explore the association between PPH and selected chronic health outcomes, for instance, CVD. Study designs could be improved by using standardised outcome assessment such as applying standardised self-report tools validated for this population or clinical interviews. Additionally, emphasis on the potential cumulative impact of PPH throughout the obstetric history of women should be considered. This is particularly important in multiparous women who may have more than one primary PPH with a potentially more severe impact. As evidenced by qualitative studies, differences in quality of health care (e.g., frequency of follow up, access to counselling and psychologist) should be also considered in any future prospective cohort studies.

In spite of increasing trends in PPH, women are less likely to experience adverse fatal outcomes due to better management and recording of severe PPH compared to the past decade [4, 87]. Globally, the research focus on other near-miss complications has shifted from maternal mortality to surveillance of the burden of severe maternal morbidity, including improvement in the quality of follow-up care [30, 88]. The same principle should be applied to future research on PPH to explore whether the trajectory of PPH and its acute morbidities is associated with subsequent risks of developing chronic disease.

More research should also be conducted to explore the potential longer-term psychological implications for partners witnessing PPH. Qualitative evidence in this review suggested signs

of acute distress, with significant information needs and follow up support needs for partners who witnessed women experiencing PPH. Obtaining information about partners' mental health and coping strategies could be beneficial to both the woman and her partner in terms of identifying support needs moving forward.

## Conclusions

This review synthesized evidence about longer-term physical and psychological health outcomes among women who had a primary PPH in high-income countries, and their partners. While the available evidence mainly focuses on immediate health outcomes following birth, this review suggests that women can have longer-term health problems associated with PPH beyond one-year after childbirth. The extent of the impact of these health outcomes is poorly researched and may be influenced by the severity of PPH, presence of other obstetric complications, and the quality of care received. The limitations of the evidence about longer-term health outcomes after PPH emphasizes the need for further research in this area.

## Supporting information

**S1 Checklist. PRISMA 2020 checklist.**
(DOCX)

**S1 File. Medline search strategy.**
(DOCX)

**S2 File. Data extraction forms.**
(DOCX)

**S1 Table. Modified risk of bias assessment tool for non-randomized studies (ROBANS).**
(DOCX)

## Acknowledgments

We would like to express our sincere appreciation to Nia Roberts (Senior librarian, University of Oxford Bodleian Health Care Libraries) for her guidance in developing search strategies.

## Author Contributions

**Conceptualization:** Rachel Rowe.

**Data curation:** Madeline Elkington, Mahkawnghta Awng Shar.

**Formal analysis:** Su Mon Latt, Madeline Elkington, Mahkawnghta Awng Shar.

**Methodology:** Su Mon Latt, Fiona Alderdice, Jennifer J. Kurinczuk, Rachel Rowe.

**Project administration:** Su Mon Latt.

**Supervision:** Fiona Alderdice, Jennifer J. Kurinczuk, Rachel Rowe.

**Validation:** Fiona Alderdice, Madeline Elkington, Rachel Rowe.

**Writing – original draft:** Su Mon Latt.

**Writing – review & editing:** Su Mon Latt, Fiona Alderdice, Jennifer J. Kurinczuk, Rachel Rowe.

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
