## [Decision Letter · Decision Letter 0]

27 Oct 2022

PONE-D-22-23242Primary postpartum haemorrhage and longer-term physical, psychological, and psychosocial health outcomes for women and their partners: a mixed-methods systematic reviewPLOS ONE

Dear Dr. Latt,

Thank you for submitting your manuscript to PLOS ONE. After careful consideration, we feel that it has merit but does not fully meet PLOS ONE’s publication criteria as it currently stands. Therefore, we invite you to submit a revised version of the manuscript that addresses the points raised by the two reviewers during the review process. 

We look forward to receiving your revised manuscript.

Kind regards,

Anayda Portela

Academic Editor

PLOS ONE

Journal Requirements:

Reviewers' comments:

Reviewer's Responses to Questions

**Comments to the Author**

1. Is the manuscript technically sound, and do the data support the conclusions?

Reviewer #1: Yes

Reviewer #2: Yes

2. Has the statistical analysis been performed appropriately and rigorously? 

Reviewer #1: Yes

Reviewer #2: N/A

3. Have the authors made all data underlying the findings in their manuscript fully available?

Reviewer #1: Yes

Reviewer #2: Yes

4. Is the manuscript presented in an intelligible fashion and written in standard English?

Reviewer #1: Yes

Reviewer #2: Yes

5. Review Comments to the Author

Reviewer #1: This systematic review of quantitative and qualitative studies on the long-term outcomes of postpartum haemorrhage is well written.

Two limitations of the review are: firstly, it includes only publications in English and secondly it includes only publications from high income countries. Globally postpartum haemorrhage is a more significant cause of maternal mortality in low- and middle-income countries which contribute to a greater share of the global burden of maternal morbidity among survivors.

The authors should explain the rationale for excluding publications from low- and middle-income countries in their review. I did not find country status as a term in the search strategy file (S1 File).

Furthermore, the authors should consider revising the title to “Primary postpartum haemorrhage and longer-term physical, psychological, and psychosocial health outcomes for women and their partners in high income countries: a mixed-methods systematic review”.

Reviewer #2: Summary of the research and overall impression

Post-partum hemorrhage is one of the main complications of childbirth worldwide and most research have been focusing on its immediate outcomes on women. This original research aims at summarizing the literature on long-term outcomes of post-partum hemorrhage (PPH) in high-income countries, on women and their partners; an objective we consider to be important for both clinical settings and policy implications. The main objective is coherent, the methods (mix-methods systematic review) is relevant, and the expected results can be significant for maternal health in high income countries. Considering women’s experience of PPH and the long-term outcomes of the partner are also strong aspects of this review.

In the article, the authors have properly placed the claims in the context of the previous literature and clearly explained why this review is needed in view of the current knowledge on the subject. Yet, we think the rationale could be emphasized and could further describe why it is important to obtain data on long-term outcomes following a post-partum hemorrhage (PPH). Indeed, the problem is well defined but does not describe the importance of getting such results. Besides, we think that exploring women’s experience of PPH is crucial and is a strength of this article but the part on experiences of care does not directly answer to the main objective of this review. Indeed, we consider that this part of the study (especially the experiences of care – see inclusion criteria for qualitative studies / outcome) answers to a different and specific objective than synthesizing the evidence about the long-term outcomes of PPH, and we recommend a clarification on this matter. Moreover, the systematic review focuses on longer-term outcomes of PPH with the main objective being to “synthesize the evidence about the long-term (…) consequences of PPH”. Yet, the inclusion criteria presented in table 1 consider that studies could be included with any follow-up duration after hospital discharge following PPH. Could we consider outcomes occurring at hospital discharge to be long-term outcomes (see table 4 | study 33 – Drayton el al. 2016 and study 32 – Chessman et al 2018)?

The protocol of the review is available on Prospero. Some minor changes from the initial protocol have been observed. Indeed, the protocol mentioned that “There will be no limitations on language” but the final manuscript states as exclusion criteria “Papers which were not written in English”. Moreover, the protocol mentioned “The search for unpublished studies and grey literature will include reports from WHO, UNFPA, NHS Maternity statistics. » but the article mentions no grey literature search nor reports from WHO, UNFPA, NHS. These changes have not been explained in the final article. Only a sentence in the conclusion explains that five articles written in another language from English have been excluded due to lack of sufficient funding. Yet, these changes are minors, and we did not observe any major deviation from the initial protocol published. The mix-method of this systematic review brings substantial value to the overall results. Some choices in the search strategy are raising questions (see below) but don’t seem to compromise the validity of the results nor induce any publication bias. The risk of bias and quality assessment is coherent and well conducted and exposed. The results and discussion parts are very clear, well-structured, and organized.

Finally, the manuscript is well organized and mostly clearly written. The introduction could benefit from revision in the writing, to clarify the link between ideas. The inclusion / exclusion criteria are clearly presented in a table (table 1) although it could be interesting to clarify some minor aspects (see below). The study is mostly conformed to PRISMA guidelines, except from minor details presented below. The tables are well presented and help the reader follow the different steps of the review: they are very clear. No plagiarism detected. No conflict of interest stated nor detected.

As is, we advise the editor to accept the article with major revision. We strongly encourage the authors to revise the few changes proposed and re-submit a revised version of their article as we consider this is solid work that just needs to be consolidated to gain clarity.

This article follows all PLOS ONE criteria for publication:

The study presents the results of primary scientific research.

Yes

Results reported have not been published elsewhere.

No

Experiments, statistics, & other analyses are performed to a high technical standard & are described in sufficient detail.

Yes

Conclusions are presented in an appropriate fashion and are supported by the data.

Yes

The article is presented in an intelligible fashion and is written in standard English.

Yes

The research meets all applicable standards for the ethics of experimentation and research integrity.

Yes

The article adheres to appropriate reporting guidelines and community standards for data availability.

Yes

Evidence and examples

Below, you will find comments for each part of the manuscript. PRIMA checklists were used to guide the reviewing process.

Major changes

The rationale / justification of this article does not emphasize enough on the importance of providing data on the long-term outcomes of PPH: why is it important in terms of clinical practice and policy implications?

o Lines 46-65: The introduction could be more developed. The rationale clearly explains that a knowledge gap exists and needs to be filled in this area of research, in view of the current knowledge on the matter (lines 51-61) but it fails to explain why it is important to identify what the long-term outcomes of PPH are, in terms of clinical practice but also in terms of policy implications.

o Moreover, it is not clear whether the studies mentioned (ref 14-18) were investigating the longer-term outcomes of PPH or the immediate outcomes.

Exploring women’s experience of PPH is key and including qualitative studies to this review is clearly a strength of the article. Yet, the experiences of care described in the eligibility criteria for the qualitative studies are more factors that could influence the occurrence of long-term consequences and ipso facto, do not answer directly to the main objective.

o The main objective of the review was to “synthesize the evidence about the long-term (…) consequences of PPH” but the paragraph on “Experiences of care” is not directly answering to this objective. It is an interesting part to understand the factors that could influence the occurrence of long-term consequences though but answers to a secondary objective.

Minor changes and questions

Abstract

• Psychosocial impacts are mentioned in the title of the review but missing in the objectives of the abstract.

• In the methods section of the abstract (see PRISMA checklist for abstracts):

o lack of mention of “systematic” review

o lack description of eligibility criteria: do not specify inclusion criteria for the review

o lack of last date of the search and description of databases used

o lack of the method used to assess the risk of bias of studies included

o lack of the method of synthesis of the results

o lack of a sentence about a summary of the limitations of the evidence included

Introduction

The writing of the three first sentences of the introduction (lines 46-51) lacks clarity in terms of linkages between ideas and the first sentences sometimes appear to have to no connection between one another. Concerning the writing, the linkage between ideas / sentences could be reinforced. Until reaching the paragraph on existing studies on the matter, it seems that the authors are presenting a succession of ideas without clear link between one another (definition – incidence – outcomes… but no written connection between these sentences). For example: Lines 48-49 about the incidence: how does it relate to the previous and following sentences? Not clear

The last sentence of the introduction is rather presenting the method to answer to the objective than presenting a clear research objective. Could be rephrased with more clarity. The last sentence of the introduction should clearly present the objective of this systematic review but here, instead, the authors are already presenting the methods used.

Material and methods

Eligibility criteria:

- Table 1 is very helpful and very clear. Yet, the reader could benefit from two small clarifications:

o The rationale for the definition of the exposure which is missing. How did the authors define the exposure definition of primary PPH? Could be interesting to explain further.

o For exposure definition of primary PPH, did the author consider the way blood loss was estimated? Not clearly stated in the eligibility criteria.

o How did the author make sure that women had really suffered from PPH in qualitative studies without any eligibility criteria for exposure?

o While reading the inclusion criteria for the outcomes, we need the information from the “time” cells to understand what “non-immediate” outcomes are. Merging the two lines of outcomes and time could help clarify this.

- The systematic review focuses on longer-term outcomes of PPH with the main objective being to “synthesize the evidence about the long-term (…) consequences of PPH”. Yet, the inclusion criteria presented in table 1 consider that studies could be included with any follow-up duration after hospital discharge following PPH. Could we consider outcomes occurring at hospital discharge to be long-term outcomes (see table 4: study 33 – Drayton el al. 2016 and study 32 – Chessman et al 2018)?

S1 File:

What motivated this search strategy?

- Why not search all the synonyms of post-natal (OR postnatal OR peri-part OR peripart…) first, and then all the synonyms of hemorrhage and then link the two searches with AND?

- Why search #20 till #38 and not apply filters? And why search these words only in title and abstract?

- Why not include the outcomes in the search strategy?

Results and discussion

- Table 2: Would be interesting to add the definition used for PPH in each study the follow-up time after PPH.

- Table 4:

• Add a column with what has been studied could be helpful (PPH? EPH? Transfusion?)

• Error in the hemoglobin unit (study 32 – Chessman 2018) -> should be >9 g/dL or 7-9 g/dL or <7 g/dL and not >90 g/dL or 70-90 or <70 g/dL

• Revise column outcomes for psychological health outcomes (missing separation between some outcomes in the table).

- For long term outcomes, no indication if patients had benefitted from therapy or not. Was it mentioned in some studies? It could really have an impact on the results.

- “It is important to provide women who had experienced PPH a thorough explanation and clear communication about their condition, treatments received including hysterectomy and its potential consequences.” (l371-373): Is it a result from the studies retrieved or a comment that should be included in the discussion part? Unclear in the way it is formulated. Should be stated from the perspective of women, from the eligible studies included in the analysis.

6. PLOS authors have the option to publish the peer review history of their article (what does this mean?). If published, this will include your full peer review and any attached files.

Reviewer #1: No

Reviewer #2: No

---

## [Author Response · Author response to Decision Letter 0]

8 Dec 2022

Thank you for taking time to review our manuscript and the detailed feedback provided. We have submitted the revised manuscript with tracked changes along with the detailed "response to reviewers' document". Please feel free to come back to us if there is any additional information needed.

Warm regards,

Su Mon Latt

---

## [Decision Letter · Decision Letter 1]

31 May 2023

Primary postpartum haemorrhage and longer-term physical, psychological, and psychosocial health outcomes for women and their partners in high income countries: a mixed-methods systematic review

PONE-D-22-23242R1

Dear Dr. Latt,

We’re pleased to inform you that your manuscript has been judged scientifically suitable for publication and will be formally accepted for publication once it meets all outstanding technical requirements.

Kind regards,

Dario Ummarino, PhD

Senior Editor

PLOS ONE

Additional Editor Comments (optional):

Reviewers' comments:

Reviewer's Responses to Questions

**Comments to the Author**

1. If the authors have adequately addressed your comments raised in a previous round of review and you feel that this manuscript is now acceptable for publication, you may indicate that here to bypass the “Comments to the Author” section, enter your conflict of interest statement in the “Confidential to Editor” section, and submit your "Accept" recommendation.

Reviewer #1: All comments have been addressed

Reviewer #2: All comments have been addressed

2. Is the manuscript technically sound, and do the data support the conclusions?

Reviewer #1: Yes

Reviewer #2: Yes

3. Has the statistical analysis been performed appropriately and rigorously? 

Reviewer #1: Yes

Reviewer #2: N/A

4. Have the authors made all data underlying the findings in their manuscript fully available?

Reviewer #1: Yes

Reviewer #2: Yes

5. Is the manuscript presented in an intelligible fashion and written in standard English?

Reviewer #1: Yes

Reviewer #2: Yes

6. Review Comments to the Author

Reviewer #1: (No Response)

Reviewer #2: Much clearer manuscript after thorough revision. We had previously advised to possibly add the exploration of the experience of care as a secondary objective but, after reading this new submission, we recommend focusing only about the long-term consequences of PPH in this review and not consider the experience of care. Indeed, if the rationale stated in the introduction is understandable, it would have requested a secondary objective applicable also for quantitative studies. We therefore recommend focusing only on PPH long-term consequences - mix-methods still. Experience of care can still be extracted from the articles and presented in the results and discussed but not mentioned from the beginning in introduction / objective / methods section. The main objective of including qualitative studies being to add value to the quantitative results by including women and their partners' perception of long-term outcomes (to be added in introduction). Exploring the association between experience of care and long-term outcomes of PPH could be the objective of another publication.

7. PLOS authors have the option to publish the peer review history of their article (what does this mean?). If published, this will include your full peer review and any attached files.

Reviewer #1: No

Reviewer #2: No

---

## [Editor Report · Acceptance letter]

2 Jun 2023

PONE-D-22-23242R1 

Primary postpartum haemorrhage and longer-term physical, psychological, and psychosocial health outcomes for women and their partners in high income countries: a mixed-methods systematic review 

Dear Dr. Latt:

I'm pleased to inform you that your manuscript has been deemed suitable for publication in PLOS ONE. Congratulations! Your manuscript is now with our production department. 

Kind regards, 

on behalf of

Dr Dario Ummarino, PhD 

Staff Editor

PLOS ONE